# Assessment of Salt Stress to *Arabidopsis* Based on the Detection of Hydrogen Peroxide Released by Leaves Using an Electrochemical Sensor

**DOI:** 10.3390/ijms232012502

**Published:** 2022-10-18

**Authors:** Jiancheng Zhang, Mei Lu, Han Zhou, Xihua Du, Xin Du

**Affiliations:** College of Life Sciences, Shandong Normal University, Jinan 250014, China

**Keywords:** salt stress, hydrogen peroxide, electrochemical sensor, nanomaterial, plant leaves

## Abstract

Salt stress will have a serious inhibitory effect on various metabolic processes of plant cells, this will lead to the excessive accumulation of reactive oxygen species (ROS). Hydrogen peroxide (H_2_O_2_) is a type of ROS that can severely damage plant cells in large amounts. Existing methods for assessing the content of H_2_O_2_ released from leaves under salt stress will cause irreversible damage to plant leaves and are unable to detect H_2_O_2_ production in real time. In this study, on the strength of a series of physiological indicators to verify the occurrence of salt stress, an electrochemical sensor for the detection of H_2_O_2_ released from leaves under salt stress was constructed. The sensor was prepared by using multi-walled carbon nanotube-titanium carbide–palladium (MWCNT-Ti_3_C_2_T_x_-Pd) nanocomposite as substrate material and showed a linear response to H_2_O_2_ detection in the range 0.05–18 mM with a detection limit of 3.83 μM. Moreover, we measured the determination of H_2_O_2_ released from *Arabidopsis* leaves at different times of salt stress by the sensor, which was consistent with conventional method. This study demonstrates that electrochemical sensing is a desirable technology for the dynamic determination of H_2_O_2_ released by leaves and the assessment of salt stress to plants.

## 1. Introduction

At present, about one-fifth of the irrigated soil in the world is affected by salinization [1]. Under the condition of soil salinization, salt stress has caused certain economic losses to agricultural production [2,3,4]. Salt stress caused by saline alkali land has a significant impact on normal plant life activities, and high salt can induce osmotic stress, ion stress, and oxidative stress in plants [5,6,7,8,9]. In a high-salinity environment, osmotic stress occurs first in plants, then ion stress occurs later, which will destroy ion homeostasis in plants, thus affecting plant growth. Osmotic stress and ion stress can induce oxidative stress in plants and will lead to the excess production of ROS, which can cause cell damage and death. ROS are an important signaling molecule that play an important role in biotic and abiotic stress in plants. It can sense and integrate environmental signals quickly in cells in order to improve the tolerance of plants to the environment [10]. Among ROS, H_2_O_2_ is one of the most stable forms which participates in electron transfer during photosynthesis in and out of the body [11,12,13]. H_2_O_2_ also acts as a signal molecule and plays an important role in the signal transduction of plants because of its long-life span and ability to cross cell membranes [14,15,16].

The traditional determination method of H_2_O_2_ released by plants is to dye plant tissues or roots with diaminobenzidine tetrahydrochloride (DAB) which can combine with H_2_O_2_ to form insoluble brown precipitation [17]. However, this method can neither achieve real-time monitoring of H_2_O_2_ release from leaves nor detect the specific amount of H_2_O_2_, and it can also cause irreversible damage to plant leaves. The method of chemiluminescence has been used to quantify the amount of H_2_O_2_ in plant tissues [18], but it is not convenient to use because it requires luminescent detection equipment to monitor the reaction. Scopoletin and 7-hydroxy-6-methoxy-2H-1-benzopyran-2-one are fluorescent substrates for peroxides and have also been used to determine H_2_O_2_ concentrations [19]. However, it is influenced by many factors and its sensitivity is relatively low. The H_2_O_2_ content is measured with spectrophotometry using phenol red as an H_2_O_2_ indicator dye. However, this method requires the fragmentation of plant tissues, which causes irreversible damage to the plants [20]. At present, non-destructive probe techniques are becoming more and more popular in the detection of H_2_O_2_ released by plants, such as Amplex Red, europium-tetracycline complex, and other probes [21]. However, these probes have certain toxic effects on plants. Nowadays, analytical methods based on electrochemical sensors have attracted widespread concern in the analysis of plants owing to its sensitivity, in situ detection ability, and so on [22].

In the process of the electrochemical sensor preparation, the selection of electrode substrate material is the key factor affecting the conductivity of the sensor. With the deepening of nanomaterials research, the application of enzyme-free H_2_O_2_ electrochemical sensors provides a new idea for researchers [23]. MWCNTs stand out among many nanomaterials with enzyme-like electrocatalytic activity because of their significant structural diversity and stable physical and chemical properties [24]. In addition, the electrochemical properties of MWCNT nanocomposite are often better than each component due to the synergistic effect [25,26]. As a new two-dimensional material, Ti_3_C_2_T_x_ shows excellent electrochemical ability because of its good conductivity and catalytic activity [27,28,29]. At the same time, Ti_3_C_2_T_x_ shows excellent catalytic ability for H_2_O_2_. At present, there have been a lot of research on electrochemical sensors constructed with Ti_3_C_2_T_x_ to further detect H_2_O_2_, and it has shown good detection performance [30,31]. Among several metal nanomaterials, Pd nanoparticles exhibit excellent electrocatalytic performance for hydrogen peroxide, and Pd further exhibits enhanced electron transfer and reduced overpotential properties [32,33,34]. Furthermore, the abundance of Pd makes it an inexpensive alternative used in a variety of fields compared to other novel metals [35].

In this work, an electrochemical sensor based on MWCNT-Ti_3_C_2_T_x_-Pd nanocomposite was constructed to detect H_2_O_2_ released from *Arabidopsis* leaves and aim to assess salt stress (Figure 1). The electrochemical information of H_2_O_2_ produced by *Arabidopsis* leaves under salt stress was compared with the results obtained by conventional staining. Compared with current methods, samples of leaves would not to be lysed or fixed, which maintain the living and original forms of biological samples. At same time, the sensor can detect the release of H_2_O_2_ in different time periods after salt stress to measure the real-time dynamic detection of H_2_O_2_ release from leaves. This electrochemical method provides a certain basis for the detection of harmful substances produced by plants under other stress conditions.

## 2. Results and Discussions

### 2.1. Characterization of the MWCNT-Ti_3_C_2_T_x_-Pd Nanocomposite

To improve the performance of the electrochemical sensor, MWCNT-Ti_3_C_2_T_x_-Pd nanocomposite was synthesized to enhance the signal of the electrode. In order to explore the synthesis of nanocomposite, the shape and composition of the material were characterized with a transmission electron microscope (TEM). Figure 1A showed that the MWCNTs were fibrous and their size could reach the micron level. As can be seen from Figure 1B, the Ti_3_C_2_T_x_ presented multilayer flake, which effectively expanded the electroactive area and provided more attachment sites for Pd nanoparticles. From Figure 1C, it can be seen that MWCNTs and Ti_3_C_2_T_x_ can be combined together through π-π interactions. The image showed that the MWCNT-Ti_3_C_2_T_x_ nanocomposite was successfully synthesized. At present, studies have shown that palladium nanoparticles have a significant catalytic effect on H_2_O_2_ [34]. Therefore, we reduced PdCl_2_ to Pd nanoparticles on the surface of MWCNT-Ti_3_C_2_T_x_ based on the redox method. It can be seen from Figure 1D that Pd was well adsorbed on the Ti_3_C_2_T_x_. In addition, the energy dispersive X-ray spectroscopy measurement result of the MWCNT-Ti_3_C_2_T_x_-Pd nanocomposite was shown in Figure 1E, which confirmed the existence of Pd and Ti elements in the nanocomposite. These results demonstrated the successful synthesis of the MWCNT-Ti_3_C_2_T_x_-Pd nanocomposite.

### 2.2. Electrochemical Characterizations of the Modified Electrodes 

CV is a method that is commonly used to explore electrochemical properties of molecules adsorbed on electrode surfaces. Figure 2A showed the conductivity of Ti_3_C_2_T_x_, MWCNT, MWCNT-Ti_3_C_2_T_x_, and MWCNT-Ti_3_C_2_T_x_-Pd in the 10 mM K_3_Fe(CN)_6_ solution with 0.1 M KCl. The response signal of the MWCNT-Ti_3_C_2_T_x_-Pd nanocomposite modified electrode indicated that its electron transfer speed was fast. As shown in Figure 2B, the current response of Ti_3_C_2_T_x_ nanocomposite modified electrode was relatively low. This result also showed that Ti_3_C_2_T_x_ had a poor electron transfer ability when it existed alone. The performance of MWCNTs and MWCNT-Ti_3_C_2_T_x_ nanocomposite modified electrodes were better than that of Ti_3_C_2_T_x_ modified electrodes. Moreover, the closed area of the CV curve of MWCNT-Ti_3_C_2_T_x_-Pd nanocomposite modified electrode was large, which further showed that the active surface area of the electrode was expanded, and the sensitivity of the sensor was greatly improved. Figure 2B showed that the reduction potential of H_2_O_2_ was approximately 0 V, and the MWCNT-Ti_3_C_2_T_x_-Pd nanocomposite modified electrode exhibited an excellent detection effect on H_2_O_2_ solution. This result indicated that Pd nanoparticles had an efficient catalytic effect on H_2_O_2_ [36].

This electrode-modified nanomaterial MWCNT-Ti_3_C_2_T_x_-Pd was further explored for the kinetic properties of electrochemical sensors by using the CV method. This procedure was performed by measuring the response signals at different scan rates using MWCNT-Ti_3_C_2_T_x_-Pd modified electrodes in 5 mM H_2_O_2_. As can be seen from Figure 2C, the response signal of this sensor gradually increased with the increase of the scan rate within the 20 mV/s–190 mV/s scan rate. According to the above exploration, we fit and analyze the reduction peak currents (I_p_) and the square root of the scan rate (v^1/2^) at different scan rates. It was found that the current value was directly proportional to the square root of the scanning rate, and its linear equation was I_p_ = −11.28 v^1/2^–62.28 (R^2^ = 0.998). This result further showed that the electrochemical behavior of H_2_O_2_ on MWCNT-Ti_3_C_2_T_x_-Pd nanocomposite modified electrodes was a diffusion-controlled process.

### 2.3. Optimization of Experimental Conditions

In order to enhance the conductivity of the electrochemical sensor, the dispersion effect of the nanocomposite is one of the most important factors to be considered. Therefore, MWCNTs were dispersed with water, PBS, PVP (2 mg/mL), 5% PDDA, and 98% DMF as dispersants, respectively. As can be seen from Figure 3A, MWCNTs dispersed by water and PBS had a poor dispersion effect compared with the other three dispersants. In order to further explore the dispersion properties of PVP, PDDA, and DMF, the dispersion effects of these three dispersants were evaluated with CV. Figure 3B showed that MWCNTs dispersed by 5% PDDA had the best conductive effect. Therefore, 5% PDDA was used as dispersant for subsequent experiments. The ratio between MWCNTs and Ti_3_C_2_T_x_ are also an important factor affecting the performance of the electrochemical sensor. According to previous experiments, when the concentration of MWCNTs exceed 4 mg/mL, it was not more stable on the electrode surface. The optimization of material proportion was also explored in 5 mM H_2_O_2_ solution using the CV method. It can be seen from Figure 3C that when the ratio of MWCNTs to Ti_3_C_2_T_x_ reached 4:1 (4 mg MWCNT and 1 mg Ti_3_C_2_T_x_ dissolved in 1 mL system), the sensor showed a better catalytic effect on H_2_O_2_ solution. Therefore, this proportion was used for follow-up research.

### 2.4. Amperometric Measurements of H_2_O_2_

The electrocatalytic behavior of MWCNT-Ti_3_C_2_T_x_-Pd nanocomposite for H_2_O_2_ reduction was studied using the CV method. Figure 4 showed that the MWCNT-Ti_3_C_2_T_x_-Pd nanocomposite modified electrode was placed in H_2_O_2_ solution with concentrations of 0, 1 mM, 5 mM, and 10 mM, respectively, to electrochemical detection using the CV method. As can be seen from Figure 4, with the increase of the H_2_O_2_ concentration, the current response signal of the sensor also increased gradually, and this result further showed that the potential of H_2_O_2_ reduction is the 0 V. Figure 4 showed that H_2_O_2_ can be easily electrochemically reduced on MWCNT-Ti_3_C_2_T_x_-Pd nanocomposite in a wide concentration range, which also showed that the H_2_O_2_ electrochemical sensor had a good performance.

In order to further detect the performance of a MWCNT-Ti_3_C_2_T_x_-Pd nanocomposite modified electrode to H_2_O_2_, the response ability of the electrochemical sensor to H_2_O_2_ was measured when the polarization voltage was 0 V. At the same time, in order to make H_2_O_2_ disperse evenly in the system quickly, the solution was stirred with a magnetic stirrer. The response of the electrochemical sensor to different concentrations of H_2_O_2_ was further evaluated by chronoamperometry. It was obvious from Figure 5A that the current response signal also increased gradually with the increase of the added H_2_O_2_ in the same time period. It was obvious that the MWCNT-Ti_3_C_2_T_x_-Pd nanocomposite modified electrode reacted rapidly to the change of H_2_O_2_ concentration, and the electrochemical sensor also responded well to the small concentration of H_2_O_2_. As shown in Figure 5B, the linear fitting between the concentration of the detected H_2_O_2_ solution and the corresponding current value was calculated. The linear equation of the modified electrode was y = −20.76 × −20.85 from 0.05 mM to 18 mM, and the linear correlation coefficient R^2^ = 0.993. According to the linear regression equation, the sensitivity was calculated to be 293.85 µA mM^−1^ cm^−2^, and the minimum detection limit was calculated to be 3.83 µM (LOD = 3SD/slope) based on the linear regression curve at a relatively low concentration range. Meanwhile, it could be clearly seen from Table 1 that the electrochemical sensor constructed in this study had excellent sensitivity compared with the H_2_O_2_ sensor reported in previous studies. These results further showed that MWCNT-Ti_3_C_2_T_x_-Pd nanocomposite was an excellent material for the detection of H_2_O_2_, which also greatly improved the performance of the electrochemical sensor.

### 2.5. Stability and Reproducibility

In the actual monitoring process, the electrode will be affected by the external environment. Therefore, the stability and reproducibility of the nanocomposite modified electrode becomes very critical for the performance of the electrochemical sensor. The stability experiment of the material was carried out by placing the MWCNT-T_i_C_2_T_x_-Pd modified electrode in a 5 mM H_2_O_2_ solution and using the CV method to explore it. The modified electrodes were measured every two days and stored in a 4 °C environment for the next measurement after each measurement. As can be seen from Figure 6A, the electrodes still reacted well to H_2_O_2_ on the sixth day. This result showed that the electrochemical sensor exhibited good stability, which laid a foundation for the long-term determination of H_2_O_2_ released from leaves.

For reproducibility studies, the current measurements of 5 mM H_2_O_2_ were repeated five times in PBS as shown in Figure 6B. The responses of the five electrodes to H_2_O_2_ detection showed acceptable deviation, and the residual standard deviation was calculated to be 6.23%. Therefore, the MWCNT-T_i_C_2_T_x_-Pd nanocomposite modified electrode displayed a good reproducibility and considerable stability, which could be used in the detection of H_2_O_2_ released by plant leaves.

### 2.6. Physiological Characteristics of Salt Stress to Arabidopsis Leaves

Wild-type *Arabidopsis* plants with the same growth vigor were selected and divided into a control group and salt stress group. After the salt stress group was treated with 100 mM NaCl for seven days, the physiological characteristics of *Arabidopsis* were determined to verify the salt stress. Compared with the control group, the fresh weight of the shoots in the salt stress group was significantly reduced (Figure 7A). Under salt stress conditions, the contents of chlorophyll a, chlorophyll b, and total chlorophyll in the control group were significantly higher than those in the salt stress group (Figure 7B–D). Meanwhile, the level of MDA in the salt stress group was significantly higher than that in the control group (Figure 7E). 

Since salt stress usually leads to ionic stress, we determined the Na^+^ and K^+^ content of leaves. Under the condition of salt stress, the Na^+^ content of the salt stress group was significantly higher than that of the control group, while the K^+^ content was significantly lower than that of the control group (Figure 7F,G). In order to compare the changes of ion content more intuitively, we calculated the ratio of Na^+^ and K^+^ and found that the Na^+^/K^+^ value (salt stress group) was higher than that of the control group after salt treatment (Figure 7H). Proline is an osmotic regulator. When plants are subjected to salt stress, they will release a large amount of proline to regulate osmotic balance. As we expected, it was found that the proline level of the salt stress group was significantly higher than that of the control group (Figure 7I). The above physiological data indicated that plants were indeed damaged by salt stress.

### 2.7. Detection of H_2_O_2_ Released from Arabidopsis Leaves

In order to evaluate the amount of H_2_O_2_ released by *Arabidopsis* under salt stress, traditional DAB staining and electrochemical sensing technology were used to perform the qualitative and quantitative analysis of H_2_O_2_ released from leaves. It can be seen from the phenotypic map of markers in Figure 8A that the leaves of plants under NaCl stress turned yellow and wilted and the growth conditions were worse than those in the control group. Traditional DAB staining showed that the leaves of the salt treatment group was darker than the control group. This result demonstrated that plants under salt stress will produce a large amount of H_2_O_2_, which will cause serious damage to plants.

In order to perform the quantitative analysis of H_2_O_2_ released from *Arabidopsis* leaves, a prepared electrochemical sensor was used. Since the quality of each leaf was different, R_0_ (ratio of current to leaves mass) was used to analyze the results. Figure 8B showed that the H_2_O_2_ released by leaves also gradually increased with the increase of time, and that the H_2_O_2_ release of the salt treatment group was higher than the control group. The result was consistent with that of DAB staining. At the same time, we can calculate the amount of H_2_O_2_ released from each leaf based on the measured current value. Therefore, the dynamic quantitative monitoring of H_2_O_2_ released by leaves under salt stress was achieved with an electrochemical method. This provides a new method for the evaluation of some markers produced by plants under salt stress.

## 3. Materials and Methods

### 3.1. Reagents and Apparatus

K_3_Fe(CN)_6_, polyvinylpyrrolidone (PVP) and phthalic diglycol diacrylate (PDDA) were purchased from Sigma (Saint Louis, MO, USA). H_2_O_2_ was purchased from Shanghai Wokai Biotechnology Co., Ltd. (Shanghai, China). The MWCNTs, Ti_3_C_2_T_X_, and PdCl_2_ were purchased from Nanjing Xianfeng Nano Technology Co., Ltd. (Nanjing, China). Potassium chloride, ethanol, sodium chloride (NaCl), sodium borohydride (NaBH_4_), hydrochloric acid (HCl), N,N-dimethylformamide (DMF), L-proline, trioxohydrindene monohydrate, glacial acetic acid, phosphoric acid, trichloroacetic acid (TCA), and thiobarbituric acid (TBA) were obtained from Sinopharm Chemical Reagent Co., Ltd. (Shanghai, China). 5-Sulfosalicylic acid was obtained from the Tianjin Fuchen Chemical Reagents Factory (Tianjin, China). Acetone was obtained from Yantai Yuandong Fine Chemicals Co. Ltd. (Yantai, China). Toluene was purchased from the Laiyang Economic and Technological Development Zone fine chemical plant. Phosphate buffer solution (PBS) was purchased from Biological Industries. DAB was sourced from Biotopped.

The morphology characterization of nanomaterials was analyzed with an HT-7800 transmission electron microscope (Hitachi Limited, Tokyo, Japan). A CHI 400C electrochemical workstation (Chen Hua Company, Shanghai, China) with a standard three-electrode configuration was used to perform electrochemical properties. Platinum wire, a Ag/AgCl electrode, and a glassy carbon electrode (GCE, diameter 3 mm) were used as the counter electrode, reference electrode, and working electrode, respectively. The absorbance value was measured with a UV-visible spectrophotometer (Beijing General Analysis Instrument Co., LTD., Beijing, China). Ion content was determined using a flame photometer (Cole Parmer Instrument Co., made in the UK, St. Neots, UK).

### 3.2. Preparation of MWCNT-Ti_3_C_2_T_X_-Pd Nanocomposite

The dispersing effect of different kinds of dispersants is evaluated by using MWCNTs. MWCNTs were dissolved with different dispersants including ultrapure water, PBS (0.1 M, pH 7.4), PVP (2 mg/mL), 98% DMF and 5% PDDA. After that, the MWCNTs were subjected to ultrasound for 2 h and stirred for 4 h at room temperature. The dispersed MWCNTs were washed by solution (ethanol: ultrapure water = 1:1) and centrifuged at 15,300 r/min for 10 min. This process was repeated 3–5 times, and finally ultrapure water was added to obtain MWCNT uniform dispersion.

The synthesis of MWCNT-Ti**_3_**C**_2_**T**_x_**-Pd nanocomposite was to add MWCNTs and Ti**_3_**C**_2_**T**_X_** to 1mL of the optimized dispersant, and then add 20 μL 0.5 mol/L PdCl**_2_** solution (dissolved in 10% HCl) and 200 μL 2.5 mg/mL NaBH**_4_** (dissolved in ultrapure water). Finally, the composites were subjected to ultrasonic (2 h), stirring (4 h), and centrifugal washing (3–5 times). After the PDDA was washed off, the composite was resuspended with ultrapure water to obtain a uniform and stable MWCNT-Ti_3_C_2_T_x_-Pd solution.

### 3.3. Fabrication of the Modified Electrode

Before modifying the electrode, the undecorated GCE was polished with alumina powder (0.3 μm to 0.05 μm) to gain a smooth surface, and then a mixed solution of ultrapure water and ethanol was used to wash the GCE to remove the physically adsorbed substances. After that, the GCE was dried with nitrogen. Subsequently, MWCNT-Ti_3_C_2_T_x_-Pd nanocomposite dispersion (10 μL) was dropped to the conductive area of the electrode, and then the electrode was dried at room temperature.

### 3.4. Electrochemical Measurements

Cyclic voltammetry (CV) was used to characterize the electrochemical performance of the electrochemical sensor. PBS was used as the electrolyte for the determination of H_2_O_2_. Different concentrations of H_2_O_2_ solution were obtained by diluting 30% H_2_O_2_ solution in PBS. The determination of H_2_O_2_ solution was carried out under the condition of stirring. Differential pulse voltammetry was used to investigate the release of H_2_O_2_ from leaves under salt stress conditions. All electrochemical measurements were made at room temperature.

### 3.5. Physiological Characteristics of Wild-Type Arabidopsis before and after Salt Treatment

Wild-type *Arabidopsis* grew in an artificial climate incubator in the Plant Physiology Laboratory, Shandong Normal University. After 27 days of normal growth for the *Arabidopsis*, the samples were treated with 100 mM NaCl for one week for subsequent physiological analysis of leaves under salt stress and electrochemical determination of the H_2_O_2_.

#### 3.5.1. Determination of the Fresh Weight

Wild-type *Arabidopsis* with the same growth was selected and divided into the control group and salt stress group. The salt stress group was treated with 100 mM NaCl for 7 days, and the fresh weight of the aboveground part of the control group and salt stress group was weighed.

#### 3.5.2. Determination of Chlorophyll Content

0.2 g fresh leaves were weighed and cut into filaments, then put into a mixture of ethanol (95%): acetone (80%) = 1:1. After 48 h of extraction in darkness, we measured the absorbance values at wavelengths of 663 nm and 645 nm, and calculated the chlorophyll a, chlorophyll b, and total chlorophyll according to its formula.

#### 3.5.3. Determination of Malondialdehyde (MDA) Content

Seedlings (0.2 g) were mixed with 2.5 mL of 0.1% TCA and ground to homogenate, then the extract was mixed with 2.5 mL of 0.5% TBA. The mixture was soaked in boiling water for 10 min and quickly cooled on ice. The supernatant was taken and its absorbance values at wavelengths 532 nm and 600 nm were measured [41].

#### 3.5.4. Determination of Na^+^, K^+^ Content and Calculation of Na^+^ to K^+^ Ratio

0.3 g leaves were weighed and bathed in boiling water for 3 h. Then Na^+^ and K^+^ were measured in a constant volume of 25 mL and calculated with a flame spectrophotometer.

#### 3.5.5. Determination of Proline Content

0.2 g leaves were weighed and ground by adding sulfosalicylic acid, then the extract was bathed in boiling water for 10 min and centrifuged. The supernatant was removed, mixed with glacial acetic acid and acidic ninhydrin solution, and bathed in boiling water for 1 h. After cooling, toluene solution was added, and the upper toluene solution was taken to measure its absorbance value at 520 nm.

### 3.6. Preparation and Analysis of Leaves Model

In the DAB staining part, the pH of the DAB dye solution (1 mg/mL) was first adjusted to 5.8 with NaOH. Secondly, the leaves were then added to the DAB solution and left to stand in the dark at 28 °C for 8 h. After standing, the DAB solution was discarded, 10 mL of 80% ethanol was added, and boiled in water for 5 min. Finally, the waste liquid was discarded, 10 mL of absolute ethanol was added, and water was added and boiled until the leaves turned white. The samples were photographed for observation and stored in anhydrous ethanol.

During electrochemical detection, the leaves were picked from the plant, weighed, and then put into 5 mL PBS for electrochemical determination. The experiment adopted the method of differential pulse voltammetry to measure the H_2_O_2_ released from leaves at the potential of −0.3 V to 0.5 V. Meanwhile, the release of H_2_O_2_ from the leaves every two hours in 0–8 h was measured with an electrochemical sensor.

## 4. Conclusions

In this work, we prepared an electrochemical sensor based on MWCNT-Ti_3_C_2_T_x_-Pd that showed a good performance for the determination of H_2_O_2_. Based on the successful construction of the salt stress model by measuring a series of physiological indicators, we used a self-made electrochemical sensor to indirectly evaluate salt stress suffered by *Arabidopsis* thaliana by detecting the H_2_O_2_ released by the leaves. The research results showed that the content of H_2_O_2_ released by *Arabidopsis* was improved under the condition of salt stress and accumulated more with longer time. The results obtained with an electrochemical sensor were consistent with those obtained by traditional DAB staining, which further showed that the sensor had a good new performance in detecting H_2_O_2_. More importantly, the method based on electrochemical sensor overcomes the limitations of DAB staining and measures the quantitative determination of H_2_O_2_ under the minimum damage to plants. Therefore, this work provides a new method for the dynamic monitoring of hazard markers produced by plants under abiotic stress conditions.

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
