# Peer review of "Assessment of Salt Stress to Arabidopsis Based on the Detection of Hydrogen Peroxide Released by Leaves Using an Electrochemical Sensor"

_ijms, 2022, doi:10.3390/ijms232012502_

Round 1
Reviewer 1 Report
1-Please the Authors have to add the concentration of K3Fe(CN)6
2-In Figure 2D you have to add minus sign in the linear equation reported
my comments, as requested, are the following:The novelty of the manuscript proposed by the authors is the realization of H2O2 determination from Arabidopsis leaves at different time of salt stress by the sensor. This electrochemical method provides a basis for the detection of harmful substances produced by plants under other stress conditions.
The interest to the readers can be focused on environmental topic. As the authors reported in the text, about one fifth of the irrigated soil in the world is affected by salinization. Under the condition of soil salinization, salt stress has caused also economic losses to agricultural production.
The scientific characteristics could be ascribed to the detailed synthesis of MWCNT-Ti3C2TX-Pd nanocomposite as platform for the sensor proposed by the authors. Moreover, the method based on electrochemical sensor overcomes the limitations of diaminobenzidine tetrahydrochloride (DAB) staining and realizes the quantitative determination of H2O2 under the minimum damage to plants.
Author Response
Reviewer #1:
Comments:
1. Please the Authors have to add the concentration of K3Fe(CN)6.
Responses: Thank you for this suggestion to make our manuscript. We have added the concentration of K3Fe(CN)6 solution.
2. In Figure 2D you have to add minus sign in the linear equation.
Responses: Thanks for your scrutiny into our manuscript. We have made the changes as shown in Figure 2D.
Reviewer 2 Report
Author should add some more study in introduction part to make it effective.
Author should present a comparison of this study with previous findings.
Author should make conclusion more effective.
Author Response
Reviewer #2:
Comments:
1.Author should add some more study in introduction part to make it effective.
Responses: Thanks for your scrutiny into our manuscript. We have already added relevant contents.
2.Author should present a comparison of this study with previous findings.
Responses: Thank you for this suggestion to make our manuscript better. We have added the relevant content in Table 1.
3.Author should make conclusion more effective.
Responses: Thank you very much for your question. We have made changes of conclusion.
Reviewer 3 Report
This paper was an easy read as results followed nicely. Once small issues is that Ti2C2T is introduced without much discussion (page 2). I think it need a seastainces or two more to describe it. Where is the Cu coming from in the XRD?
Can you comment on the time response of the sensors? Fig 5 implies that it is seconds. The last calibration point seem off. The authors do not include the point after 18 mM or the ppoint after 23.3 mM. Is there an upper limit of H2O2?
How was the H2O2 concentration in the calibration experiment determined. I could of missed it elsewhere, so it is worth repeating.
Author Response
Reviewer #3:
Comments:
1. This paper was an easy read as results followed nicely. Once small issues are that Ti3C2Tx is introduced without much discussion (page 2). I think it need a substance or two more to describe it.
Responses: Thank you for this suggestion to make our manuscript better. We have supplemented relevant content in the conclusion as required.
2. Where is the Cu coming from in the XRD?
Responses: Thank you very much for your question. In the XRD characterization experiment, the composite needs to be dropped onto the Cu mesh for subsequent characterization, so the background peak of Cu will appear in this Fig. 1E.
3. Can you comment on the time response of the sensors? Fig 5 implies that it is seconds. The last calibration point seems off. The authors do not include the point after 18 mM or the point after 23.3 mM. Is there an upper limit of H2O2?
Responses: Thanks for your scrutiny into our manuscript. When adding different concentrations of H2O2 into the solution, the sensor can respond in 5 seconds. Meanwhile, we have adjusted Fig. 5B and added the concentration of 23.3 mM. There is a good linear relationship in the concentration range of 0.05-18 mM. The sensor has no upper limit for detecting of H2O2 but excessive concentration such as 23.3 mM, will exceed the linear range which is inaccurate.
4. How was the H2O2 concentration in the calibration experiment determined. I could have missed it elsewhere, so it is worth repeating.
Responses: Thank you very much for your question. The response of the sensor to different concentrations of H2O2 solution can be seen in Fig. 5B. In this calibration experiment, each concentration was performed in triplicate. The H2O2 concentrations in the calibration experiment were determined according to the i-t experiment as shown in Fig. 5A.